# Learning the Relative Dynamic Features for Word-Level Lipreading

**DOI:** 10.3390/s22103732

**Published:** 2022-05-13

**Authors:** Hao Li, Nurbiya Yadikar, Yali Zhu, Mutallip Mamut, Kurban Ubul

**Affiliations:** 1School of Information Science and Engineering, Xinjiang University, Urumqi 830046, China; ritianli@stu.xju.edu.cn (H.L.); nurbiya@xju.edu.cn (N.Y.); zhuyl@xju.edu.cn (Y.Z.); 2Xinjiang Key Laboratory of Multilingual Information Processing, Urumqi 830046, China; 3Technology Department, Library of Xinjiang University, Urumqi 830046, China

**Keywords:** Visual Speech Recognition, lipreading, spatial–temporal feature extraction

## Abstract

Lipreading is a technique for analyzing sequences of lip movements and then recognizing the speech content of a speaker. Limited by the structure of our vocal organs, the number of pronunciations we could make is finite, leading to problems with homophones when speaking. On the other hand, different speakers will have various lip movements for the same word. For these problems, we focused on the spatial–temporal feature extraction in word-level lipreading in this paper, and an efficient two-stream model was proposed to learn the relative dynamic information of lip motion. In this model, two different channel capacity CNN streams are used to extract static features in a single frame and dynamic information between multi-frame sequences, respectively. We explored a more effective convolution structure for each component in the front-end model and improved by about 8%. Then, according to the characteristics of the word-level lipreading dataset, we further studied the impact of the two sampling methods on the fast and slow channels. Furthermore, we discussed the influence of the fusion methods of the front-end and back-end models under the two-stream network structure. Finally, we evaluated the proposed model on two large-scale lipreading datasets and achieved a new state-of-the-art.

## 1. Introduction

In daily life, people not only communicate with others according to audio signals, but they also get the meaning of each other according to lip movements in some special scenes. In computer science, speech recognition depends on audio signals, while Visual Speech Recognition (VSR, also called lipreading) is progress that decodes the visual lip signals.

Most lipreading models consist of front-end and back-end modules. Among them, the front-end network pays more attention to the spatial feature extraction of the single frame, while the back-end network is focused more on the temporal dynamics of the whole image sequence. Limited by the structure of our vocal organs, the number of distinguishable pronunciations we could make is finite [1]. During the lipreading system, in addition to meeting some common challenges in image processing, such as imaging conditions, multiple angles, and low resolution, we also have to face the influence of homophones. Some phonemes and visemes are similar (e.g., the letter “p” and “b”) [2] and lead to similar lip motions when people say words made up of these letters; for example, the words “back” and “pack” in English, or “dumpling” and “sleep” in Chinese. On the other hand, due to the different habits of each speaker, different speakers will have different lip movements for the same word. Therefore, researchers have been concentrated on obtaining as much dynamic characterization as possible, extracting fine-grained spatial features while also obtaining temporal information of movement sequences.

From the model’s perspective, researchers have been looking for a more efficient network of back-end models for a long time. Chung et al. [3] used MT-VGG to train the end-to-end lip recognition model and found that the network structure composed entirely of 2D CNN is better than 3D CNN. Stayfylakis et al. [4] combined 2D and 3D convolution with LSTM. Specifically, a shallow 3D CNN structure is used to obtain short-term temporal features, then 2D convolution is used to extract fine-grained spatial features, and BiLSTM is used to extract temporal features from the obtained feature vectors. It is worth mentioning that the subsequent lipreading models are composed of such front-end networks and back-end networks. The front-end network composed of CNN is mainly used to obtain spatial features and then using a statistical model or RNN to model the features. Later, lipreading was mainly improved by changing the back-end network, such as B. Martinez et al. [5] used multi-scale TCN.

From the perspective out of the model, researchers have greatly improved by adding additional information without changing the structure of the model. For example, since 2D CNN cannot obtain time information, Chung et al. [3] stacked consecutive frames and sent them to 2D CNN. Similarly, Hao et al. [6] integrated the Temporal Shift Module into 2D CNN and shifted the near frames of each frame sequence so that the current frame could obtain the relevant information of the front and rear frames. Stafylakis et al. [7] introduced word boundaries to provide context information for classifying target words bypassing the boundary vector of words. On the input data, the optical flow data used to describe the instantaneous motion state of the moving object can also reflect the motion information of the lip. In [8], two-stream I3D is used to receive gray video and optical flow, and it is found that the front-end network composed entirely of 3D CNN can further improve the performance.

In addition, a training method of bimodal lipreading based on the combination of the audio-visual has become popular recently [9,10]. In simple terms, these methods take image sequences and speech information as inputs, extract visual and audio features during training, and memorize audio representations by modeling the interrelationships between audio and visual representations. In the inference stage, the visual input extracts the preserved audio representation from the preserved interrelation. Kim et al. [11] proposed a multi-modal bridging framework containing two modality-specific memories: a source-key memory and target-value memory. This enables it to complement the information of unimodal inputs with the recalled target modal information. Kim et al. [12] proposed the Multi-head Visual-audio Memory (MVM), composed of multi-head key memories for saving visual features and one value memory for saving audio knowledge. Yang et al. [13] proposed a feature disentanglement learning strategy with a linguistic module that extracts and transfers knowledge across modalities via cross-modal mutual learning.

The key point in this paper is to obtain the subtle lip motion to distinguish the similar mouth shape. Under such consideration, we proposed a new dual-stream lipreading model called Lip Slow-Fast (LSF) based on the Slow-Fast Net [14]. To obtain subtle lip motion features, two streams with different channel capacities are used to extract dynamic features and static features respectively. Compared with action recognition, lipreading is a fine-grained video multi-classification. The action could be inferred from several frames, while the lip motion is smooth and inconspicuous. That is why the performance of the Slow-Fast Net in lipreading is very poor, even far weaker than the baseline. Therefore, based on some existing studies, we discussed the front-end models of different convolutional structures and reconstructed the Slow-Fast Net, which reduces the total parameters and greatly increases the model’s performance. Furthermore, for word-level lipreading, the duration of the target word in each sample is usually less than 1s. Therefore, to better describe the whole sequence, we implemented two sampling methods and further explored their impact on the proposed model. Finally, we explored two different fusion schemes. The early fusion is for the fusion of the feature maps extracted from fast and slow channels and then sequence modeling of the relative dynamic features using Bi-Gated Recurrent Unit (BiGRU). The late fusion is to model the feature maps of two streams respectively and fuse the output results at last.

Our contributions are in four aspects:Firstly, a new front-end model with two effective CNN streams and a lateral connection is proposed in this paper to capture the relative dynamic features of lip motion.Secondly, for each component in the front-end model, we explored a more effective convolution structure and achieved an improvement of about 8%.Thirdly, we verified the impact of sampling methods due to the short duration of word-level lipreading samples on the extraction of lip motion information.Then, we discussed and analyzed the fusion methods of the two-stream front end model with the back-end model.At last, we verified our proposed method on LRW [3] and LRW-1000 [15] and achieved a new state-of-the-art.

## 2. Methods

In this section, we describe the proposed model in detail. Firstly, we introduced the structure of the front-end model, including the structural design of the residual block. Then, based on the existing research, we discussed the convolution structure of the front-end model again. Besides, we introduced two sampling methods of the proposed two-stream network and discussed the number of sampling frames in detail according to the characteristics of word-level lipreading data. Finally, we introduced the fusion method of front-end and back-end models. The overview of the proposed model is shown in Figure 1.

### 2.1. Front-End Model

In video scenes, frames usually consist of two distinct parts, a static area that does not change much or slowly and a dynamic area that is changing. For example, a video of an airplane taking off contains a relatively static airport and a dynamic object aircraft that moves rapidly through the scene. In daily life, when two people meet, the handshake is usually faster, and the rest of the scene is relatively static. Based on this insight, in the architecture of Slow-Fast Net, keyframes are processed by a deep network (slow channel) to analyze static content, and non-key frames are processed by a shallow channel capacity network (fast channel) to analyze dynamic content [16]. The data from the fast channel is then fed into the slow channel through a lateral connection, which allows the slow channel to understand the processing results of the fast channel.

Inspired by the Slow-Fast Net, the proposed model in this paper follows the structure of the fast channel and slow channel. Compared with action recognition, lipreading is a fine-grained video multi-classification. The action could be inferred from several frames, while the lip motion is smooth and inconspicuous. At present, the backbone of the front-end is usually composed of ResNet. We followed the structure of Slow-Fast Net and implemented it with ResNet-18. In addition, to extract features from low-resolution input sequences, we reduced the stride of the connection layer between the two channels. Due to the importance of spatial features, we replaced all 3D convolutions with 2D convolutions, which reduces many parameters and makes the model easier to train. In each residual block, we added a Temporal Shift Module (TSM) [6,17] and SE block to extract the temporal information better in the front-end. Figure 2 shows the working process of TSM by moving the feature map in two directions along the time dimension. Then we filled the empty channel with “0” and deleted the extra part. In the feature map after moving, the first half of the current frame gets the information of the before and after frames, and the last half retains the knowledge of the current frame. The realization of the residual block is shown in Figure 3.

### 2.2. Different Structures of Front-End

In the development of lipreading, the convolution structure of the front end can be divided into three types: full 2D CNN, full 3D CNN, and the hybrid structure of 3D and 2D CNN.

2D CNN is often used to extract static spatial information. Therefore, the Multi-Tower Visual Geometry Group (MT-VGG) [3] is wholly based on 2D CNN and each tower takes as a single frame as input. But lipreading belongs to the video recognition task, and the full 2D CNN can’t extract temporal information. Compared with 2D CNN, 3D CNN can extract short-term spatial-temporal features, but it brings much more computation and model parameters. This will make the model difficult to converge, so few lipreading models use this structure. Later, Weng et al. [8] used pre-training weights in the ImageNet dataset when training their proposed I3D model.

Nowadays, most lipreading models are based on a shallow 3D CNN and deep 2D CNN in the front-end. Under such a structure, the front-end model first obtains the short-term temporal features through the single-layer 3D CNN and then extracts the spatial features by 2D CNN. At the same time, this structure solves the problem that 3D CNN is difficult to train and 2D CNN cannot obtain temporal information. The original Slow-Fast Net used the full 3D CNN to capture dynamic information. Considering the above reasons, we show the convolution structure of each component in Table 1.

### 2.3. Back-End Model

Three-layer Bi-Gated Recurrent Units (BiGRUs) formed our back-end network with 1024 hidden units. BiGRU is composed of two unidirectional GRUs with opposite directions, and the final output is determined by both of them. Perform sequence modeling on the feature vector output by the front-end network and extract the temporal dynamics. Finally, the output is classified through a full connection layer.

### 2.4. Sampling Methods

#### 2.4.1. Interval Sampling

In Slow-Fast Net, the different sampling ratio of the slow and fast channel is also an important factor. Compared with action recognition, the action could be inferred from several frames in action recognition, while the lip motion is inconspicuous.

For word-level lipreading, the duration of each sample is within 2s, and the keyframe is even shorter. For example, Figure 4 shows a sample of the word “DEATH” in the LRW dataset. It can be observed that the range of obvious changes in lip motion is from frame 3 to frame 16. However, from frame 17 on, the speaker’s lip shape change is not obvious. Under this observation, this chapter sets the sampling interval ratio of the fast channel and the slow channel as 1–2, 1–3, 1–6, 2–6, and 1–12. This paper defines the interval sampling ratio as the number of sampling frames in the fast channel and the number of sampling frames in the slow channel. For example, for 1–2, fast channels are sampled every frame and slow channels every two.

#### 2.4.2. Sparse Difference Sampling

Another method is to obtain the differences between moving objects at different moments through the framing strategy of sparse sampling [18,19]. Specifically, each video, *V*, is first divided into *T* clips of equal duration, which do not overlap. If one frame is selected randomly from each fragment, then the total *T* frame is obtained, i.e., $I=\left(\;I_{1},\;I_{2},\;\ldots \;,\;I_{T}\right)$, and its size is *T × C × H × W*. Each segment is then subtracted from the previous segment to obtain the difference between adjacent segments. Finally, the different tensor describing the video clip is obtained by splicing the temporal dimension.

The whole process is shown in Figure 5, and the data used is the sample of the LRW dataset, of which the total frame number is 29. We separate the origin sequence into seven subsequences, and each has four frames. Then subtract them from each other to get six differential sequences. Finally, these sequences are spliced in the temporal dimension to get a 24-frame subsequence. To realize this strategy, for an odd number of frames, the last frame is removed by default so that the frame number of each fragment is the same and will not be repeated.

### 2.5. Fusion Methods

#### 2.5.1. Early Fusion

For early fusion, we first concatenated two feature vectors extracted from fast and slow channels and then sent them to BiGRUs for sequence modeling. This is to give full play to the advantages of the Slow-Fast Network and aggregate complete spatial-temporal features.

#### 2.5.2. Late Fusion

For the late fusion, our idea is to regard the Slow-Fast model as two independent stream CNN for feature extraction separately to obtain different representative spatial features through different sampling intervals. Then, we sent the obtained feature vectors into two different BiGRUs layers, respectively, and concatenated the outputs to obtain the final classification last.

## 3. Results

The proposed model was trained and evaluated on two VSR benchmarks: the Lip Reading in the Wild (LRW) dataset and the LRW-1000 dataset corpus.

In this section, we first briefly introduced the two datasets we used and presented some implementation details. Next, we conducted detailed ablation experiments of model structures, sampling methods, and fusion methods. Finally, we compared the proposed model with other relative works and made some useful remarks.

### 3.1. Datasets

LRW is a video clip extracted from the BBC TV broadcast. Each clip has 29 frames of images. It contains more than 1000 speakers and 500 English words.

The LRW-1000 includes 1000 Chinese word classes, as well as clips from TV programs that include more than 2000 speakers. Both datasets have been preassigned for training, validation, and testing. In LRW-1000, the publisher has even cropped the faces. In addition, the speakers in both datasets are diverse in posture, age, and skin color, which is the largest and most challenging lipreading dataset currently recognized. Table 2 shows the details of LRW and LRW-1000.

Figure 6 shows the processing of the training data.

### 3.2. Implementation Details

All the training and inferencing were implemented in the Linux environment with two NVIDIA RTX 2080ti. We use the Adam optimizer with the initial learning rate of 3 × 10^−4^ and the weight decay of 1 × 10^−4^. Besides, we use a cosine annealing scheduler with the max epoch of 30.

#### 3.2.1. Data Processing

Zhang et al. [20] found that face alignment is helpful for face recognition and can improve the accuracy of VSR. Firstly, we used the Dlib toolkit [21] to get landmarks of each face and then applied Procrustes analysis to gain the affine matrix. Secondly, we used a similarity transformation for all images and center cropped it with a fixed rectangle whose size is 88 × 88 to get the lip region.

In terms of data augment, mix-up and cutout are used to improve the fitting ability of the model. Mix-up is an unconventional data augment method, a simple data enhancement principle independent of data. It constructs a new training sample and its corresponding label by linear interpolation in two samples. For example, given sample *A* $\left(x_{A},y_{A}\right)$ and sample *B* $\left(x_{B},y_{B}\right)$, a new sample can be described as:(1)$$\tilde{x}=\lambda x_{A}+\left(1-\lambda \right)x_{B}\;,\;\tilde{y}=\lambda y_{A}+\left(1-\lambda \right)y_{B}\;.\;$$
where *λ* is a random parameter that obeys *β* distribution, $\lambda \sim Beta\left(\alpha ,\;\alpha \right)$, and $\left(\tilde{x}\;,\;\tilde{y}\;\right)$ is the weighted new sample. With the growing up of *α*, the training loss of the network will increase, and its generalization ability will be enhanced. In our experiment, *α* was set to 0.2.

Besides, we used the word boundary introduced by Stafylakis et al. [7] in the data processing to obtain context information. During training, the word boundary will concatenate with the feature vector exacted by the front-end and then input to the back-end to sequence modeling.

#### 3.2.2. Training Details

We introduced the label smooth to reduce over-fitting and improve the model generalization.

The label being smooth reduces the weight of the category of the real sample label in calculating the loss function and finally suppresses the over-fitting effect. Traditional cross-entropy loss is computed as:(2)$$Loss_{i}=\sum_{i=1}^{K}q_{i}logp_{i}\;,\;q_{i}=\left\{\begin{array}{c} 0,y\neq i\\ 1,\;y=i \end{array}\right.\;,$$
where *K* is the number of classes, $p_{i}$ is the prediction probability that sample *i* belongs to category *c*. $p_{i}$ is a symbolic function if the true category of sample *i* is equal to *c*, $p_{i}\;$= 1, otherwise $p_{i}\;$= 0.

After using the label smooth, the loss function is changed as:(3)$$Loss_{i}{}^{’}=\left\{\begin{array}{c} \varepsilon *Loss,\;\;\;\;\;\;\;\;\;\;\;\;y\neq i\\ \left(1-\varepsilon \right)*Loss,\;y=i \end{array}\right.\;,$$
where *ε* is a small number and was set to 0.1 in our experiment. Finally, the model makes the network have a stronger generalization ability by suppressing the output difference between positive and negative samples.

### 3.3. Ablation Experiment

We evaluated the LRW dataset according to the above experimental settings.

#### 3.3.1. The Convolution Structure of Front-End

Compared with 2D CNN, 3D can extract short-term spatial-temporal features, but it has too many parameters, and the training cost of the model is high. Besides, in recent years, most lipreading models have used 2D CNN or a mixture of 2D CNN and 3D CNN to replace 3D CNN. Based on the above researchers, we re-evaluated the structure of the whole front-end model. As shown in Table 3, the origin 3D ResNet-18 is more effective than 2D ResNet-18, and the reason is the 2D ResNet-18 cannot extract temporal information, which is important in lipreading. Then we inserted TSM into the residual block, and the performance of 2D ResNet-18 increased and was even better than 3D ResNet-18.

Besides, we also evaluated the model with three data augment methods. Cutout works by randomly covering the face with a fixed size and helps the model focus on other areas of the face, not just the lips. MixUp reduces the risk of overfitting by mixing two samples from the batch in a random proportion into a new sample. Both approaches have had great success, but as far as we know, no researchers have explored their interactions with different convolution structures. On the other hand, we introduced CutMix, which is also a very useful data augment way and can be seen as a combination of the above two methods.

Table 4 shows the effects of these three data expansion methods under different convolution structures, where C represents the proportion of covered Windows and A represents the mixing factor. Obviously, CutMix is very poor compared to cutout and mix-up. The effect of cutout in the full 2D CNN is better than that in 3D + 2D CNN. MixUp, on the contrary, performed better in 3D + 2D CNN than in the full 2D CNN.

Based on the experiment results above, we carried out experiments with the proposed model with the same sampling interval of 1–3 (fast channels sampling interval-slow channels, sampling interval). As can be seen in Table 5, when the front-end network with the full 3D structure is the structure used in the original slow-fast, it has a very poor effect. On the contrary, after replacing most of the convolution layers in the network with 2D CNN, the accuracy has been significantly improved. Although the mixture structure of 3D CNN and 2D CNN has achieved the best results, the structure of full 2D CNN also has strong competitiveness.

#### 3.3.2. Sampling Methods

The key of Slow-Fast Net is to set different sampling intervals for the fast and slow channel so that the model focuses on the dynamic region and static region, respectively. For word-level datasets, LRW and LRW-1000, the duration of each sample is within 2s, and the duration of the target word is even shorter, with the total frame number of the sample being 29–40, while the total frame number of the target word is less than 12. Therefore, for interval sampling, the sampling intervals of 1–2, 1–3, 1–6, 2–6, and 1–12 are used for this section’s experiments.

The sparse difference sampling strategy is another way to describe data. To ensure each sub-sequence is the same size and won’t be repeated, the last frame of the sample in LRW is discarded by default because the target word is usually in the middle frame so that the last frame can be regarded as redundant data. Considering the short duration of samples, we set the sparse sampling interval to 4, so that a 29-frame fragment can be divided into 7 segments, and each sub-sequence has 4 frames. In this way, the final sequence still has 24 frames after subtraction and a splicing of each segment.

The experimental results are shown in Table 6, and the method of interval sampling is obviously better than sparse difference sampling. It can be seen that when the sampling interval of the slow channel and the fast channel is 1–3, the effect is the best. When the sampling interval is 1–2, the dynamic features and static features extracted by the model are equivalent to the linear superposition of two static ones because the sampling ratio is too close. When the sampling interval is 1–6, the accuracy decreases obviously because the sampling ratio is too far; when the sampling interval is 2–6, the sampling ratio of the slow channel and the fast channel returns to 1–3, and the effect is improved to above 88% at this time. Finally, when the sampling interval is 1–12, the extreme value of lip vocalization will be missed because the fast channel has a too-large frame interval.

#### 3.3.3. Fusion Methods

As described in Section 2.5, we experimented with early fusion and late fusion structures under the same convolution structure and the same sampling interval.

The experimental results are shown in Table 7, and early fusion performs better. We believe that this problem is that the late fusion method ignores the correlation between the static and dynamic features extracted by the fast and slow channels.

#### 3.3.4. Comparison with Relative Works

The above experiments show that the Slow-Fast Network with a mixture of 3D and 2D convolution and early fusion methods with 1–3 sampling intervals has the best performance. Based on the above structure selection and parameter setting, we compared some methods in Table 8.

The Multi-Grained [22] integrates information of different granularity and learnable spatial attention modules, which can capture the nuances between words and the styles of different speakers. The two-stream network I3D [8] composed of deep 3D CNN is used to replace the previous mixture of convolution networks. For noises such as posture, speaker appearance, and speed change, GLMIN [23] combines the global mutual information constraint (GMIM) and local mutual information constraint (LMIM), enhancing the relationship between spatial information and audio information acquired. Multi-Scale Temporal Convolution Network (MS-TCN) [5] uses the temporal convolution network (TCN) instead of RNN. On the other hand, given the training sample with a temporal inconsistency problem, variable-length enhancement was proposed to solve the problem. The deformation flow network (DFN) [24] studies the deformation flow between adjacent frames. The model will extract the flow distortion and the original gray frame for combination. CBAM [20] reconsidered the key areas in lipreading and found that other regions of the face beyond the lip are also helpful. TSM [6] combined with the Temporal-Shift Module so that the current feature map could obtain the relevant information of the previous and subsequent frames by moving part channels of the current feature map in the temporal dimension.

The SpotFast [25] is also based on the Slow-Fast Net used, but our focus is completely different. To concentrate on the keyframes of the target word, the author replaced the slow channel, which uses a spot channel, and set the sampling rate of the fast channel to 1. That is, all frames are selected, which is the same as the original slow channel. At the same time, to improve the performance of the back-end model, the network places a transformer with memory augment as an encoder on each path.

Compared with these methods, the accuracy of our proposed method in LRW and LRW-1000 is 88.52% and 58.17%, respectively, which is the latest SOTA. The audio-visual method [12,13] tries to get a memory network of audio data and lip data. Among them, Kim et al. improved the previous work, which used Multi-head Visual-audio Memory (MVM) [12] and multi-temporal levels to solve homophones. They employed reconstruction-based learning and contrastive learning to train the value memory network during the training stage and improved the accuracy by 3% in LRW-1000. SE-ResNet [26] summarized the previous lipreading tricks and achieved the best performance on the two lipreading datasets, LRW and LRW-1000. Mainly applied are the word boundary and face alignment; that is the key, that the uni-modal could achieve a higher result than the bimodal. Compared with SE-ResNet, the proposed LSF in this paper could obtain more effective lip motion and improve the accuracy by 2.4% in LRW-1000.

#### 3.3.5. Statistical Results

We made statistics on the recognition results of all words in the LRW test set first and show the top-50 accuracy in Table 9. As shown in the table, there are quite a few words whose accuracy can be guaranteed to reach 100% in the complex field environment.

Figure 7 shows the effectiveness of our method for a single word in LRW. According to our statistics, there are 110 categories of words with a recognition rate of more than 95%; 212 categories of words with a recognition rate of more than 90%; 333 categories of words with a recognition rate of more than 80%; and 52 categories of words with a recognition rate of less than 50%. Compared with baseline, the number of samples whose recognition accuracy is more than 95% of LSF is doubled, while the number of samples whose recognition rate is less than 50% is reduced by 40%.

Figure 8 shows the effectiveness of our method for a single word in LRW-1000. Due to the low recognition rate of this dataset, we plotted the accuracy of each stage in detail. As shown in the figure, the improvement of LSF relative to the baseline model is mainly focused on words with a recognition rate above 80%. Although the result is not ideal, it is still the best recognition result.

## 4. Conclusions

In this paper, we focused on enhancing the ability to extract spatial-temporal features in the front-end network of lipreading. A new front-end model with two effective CNN streams is proposed to extract relative dynamic features in the image sequence. We used a larger sampling interval and a CNN stream with a lower channel capacity to capture the dynamic temporal information between sequences. For static spatial information within a single frame, we used a smaller sampling interval and a CNN stream with a larger channel capacity to capture it. At the same time, the extracted dynamic information extracted was sent to static information through lateral connection and then fused. Same as the baseline, the backbone of the proposed model is ResNet-18.

On the other hand, we combined some existing modules and data augment methods to explore the best convolution structure of the front-end model. Under the residual module designed by us, the performance of full 2D CNN with the same number of layers could achieve a similar performance to that of 3D + 2D CNN. Meanwhile, we find that the cutout is more effective for full 2D CNN, while the mix-up is more effective for 3D + 2D CNN. Considering the characteristics of word-level lipreading data, in this paper, we introduced two sampling methods and conducted detailed ablation experiments on the sampling methods. Besides, we further explored two fusion modes of the front-end net and back-end net.

Under such architecture, we verified the proposed model in two challenging lipreading datasets and demonstrated a wonderful performance. We believe the model has great potential beyond lipreading, especially in fine-grained action recognition.

In the future, we will further focus on the front-end with a lightweight structure but better performance. Besides, though the training cost of bimodal is much higher than uni-modal lipreading, the research of audio-visual lipreading is still our future work.

## Figures and Tables

**Figure 1 sensors-22-03732-f001:**
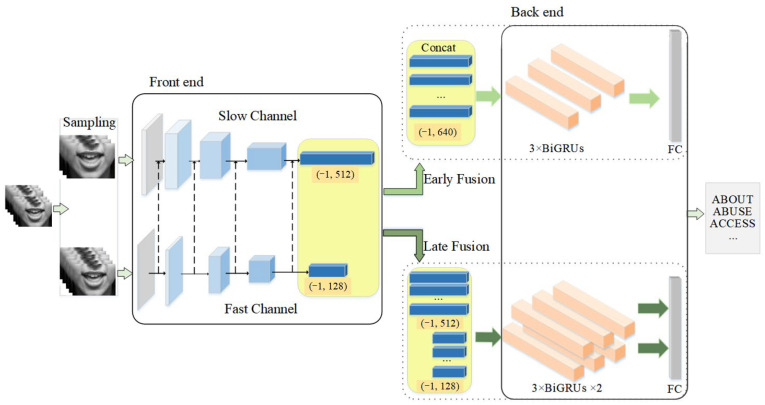
The overview of the Lip Slow-Fast (LSF). As shown in the picture above, the proposed model is a two-stream network. At first, the lip region sequence with different sampling intervals is input into two CNN streams; the above one is the slow channel, and the bottom one is the fast channel. The gray convolution is a shallow 3D CNN, and the others are 2D CNN. Then, the output feature vector will be input to the back-end model, which is a 3-layer Bi-Gated Recurrent Unit (BiGRU). In early fusion, the two feature vectors will concatenate together first and then do the sequence modeling; in late fusion, two feature vectors will send to the back-end model separately and concatenate together last. Finally, after a full connection layer, the model will predict the result of the input lip sequence.

**Figure 2 sensors-22-03732-f002:**
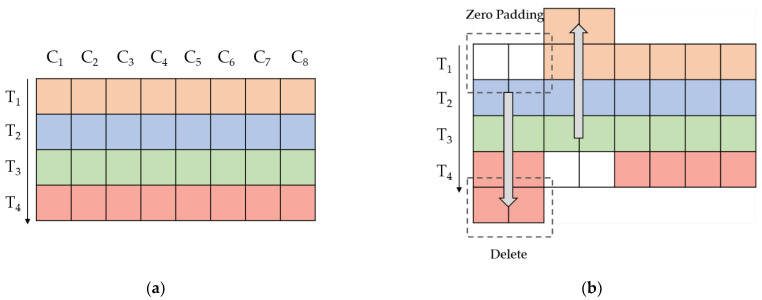
The working process of TSM. As shown in the picture, (**a**) shows a part of the characteristic diagram in the channel dimension and time dimension; (**b**) shows the working process of TSM. First, divide the feature map into two parts, moving the channel only in the first half and leaving the second half unchanged. For the moving part, the first-half channels move forward, and the last-half channels move backward for the moving part. Then, pad the empty part with “0”, and delete the extra part directly.

**Figure 3 sensors-22-03732-f003:**
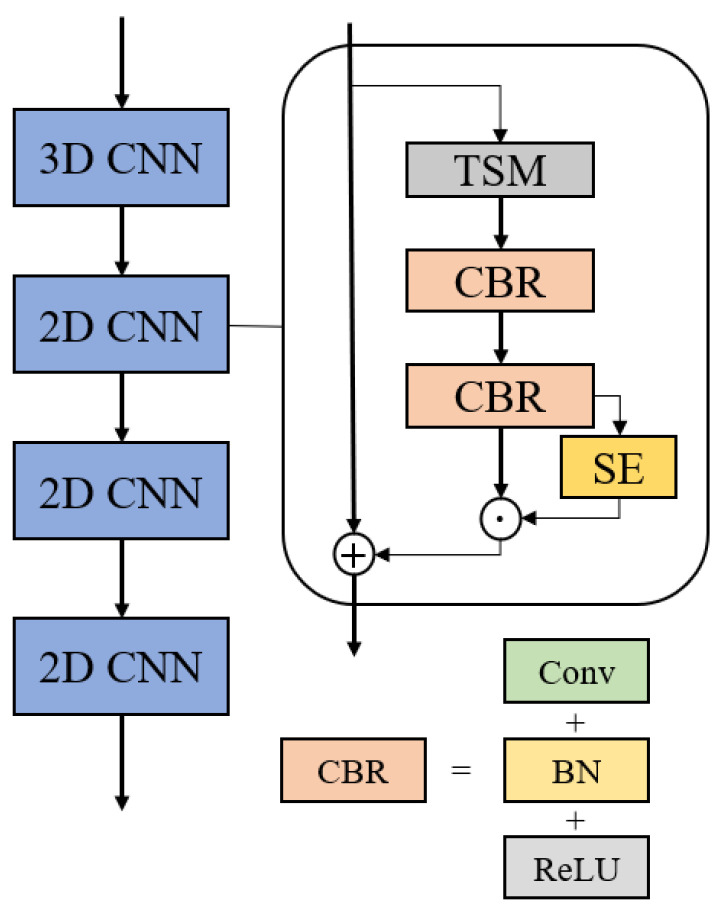
The implementation of the residual block in the proposed front-end. As shown in the picture above, in each residual block, the input data will first obtain the short-term time information through TSM. Then, go through the conventional convolution composed of two CBR. Finally, use the SE module to reweight the channel correlation.

**Figure 4 sensors-22-03732-f004:**
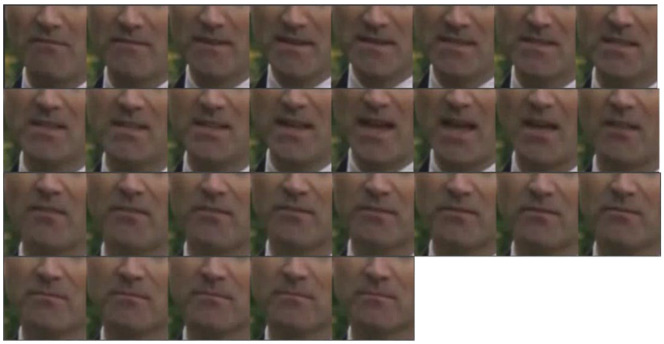
A sample of “DEATH” in LRW. It can be observed that the range of obvious changes in lip motion is from frame 3 to frame 16. From frame 17 on, the change in the speaker’s lip shape is not obvious.

**Figure 5 sensors-22-03732-f005:**
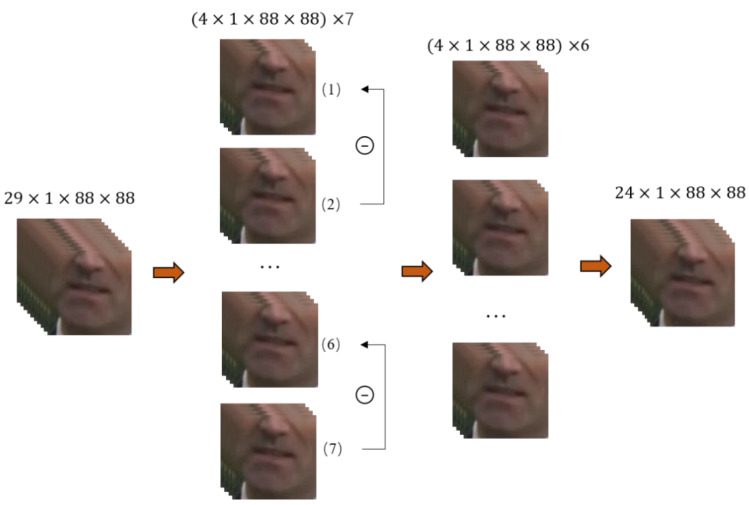
Sparse difference sampling. Given a sequence of 29 frames, we divided it into seven segments, each of which is four frames. Then subtract the previous sub-segment from the current sub-segment, for example subtracting sub-segment 1 from sub-segment 2, sub-segment 2 from sub-segment 3, etc. Finally, splice these segments in the temporal dimension to get a 24-frame result.

**Figure 6 sensors-22-03732-f006:**
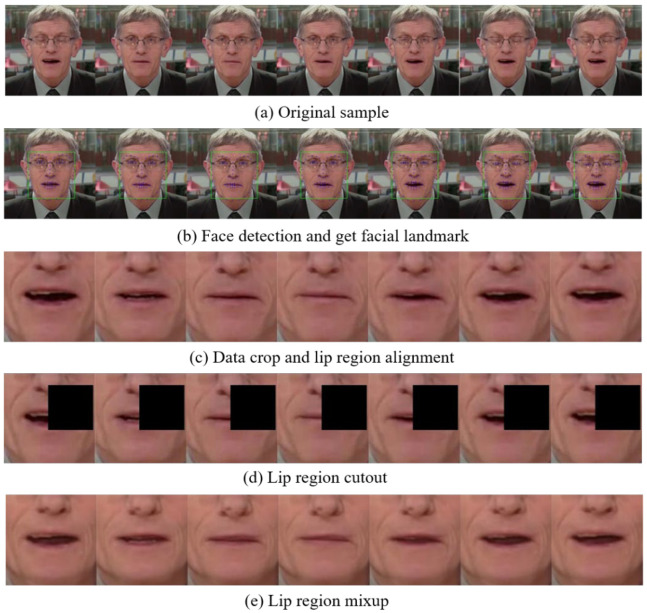
An example sample during data processing. As shown above, (**a**) is a part of the original data in the Lipreading in the Wild (LRW) dataset. (**b**) is the result of using the Dlib toolkit to get facial landmarks while the green rectangle is the region of the face, and the blue point is the facial landmarks. (**c**) is the process result of data crop and lip alignment. (**d**,**e**) is the result of mix-up and cutout during training.

**Figure 7 sensors-22-03732-f007:**
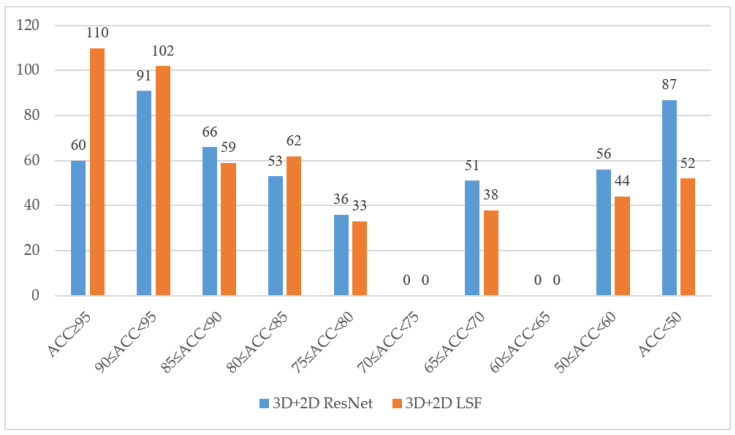
Statistics of accuracy for each class in LRW. The blue histogram is the baseline model we compared, while the orange histogram is LSF.

**Figure 8 sensors-22-03732-f008:**
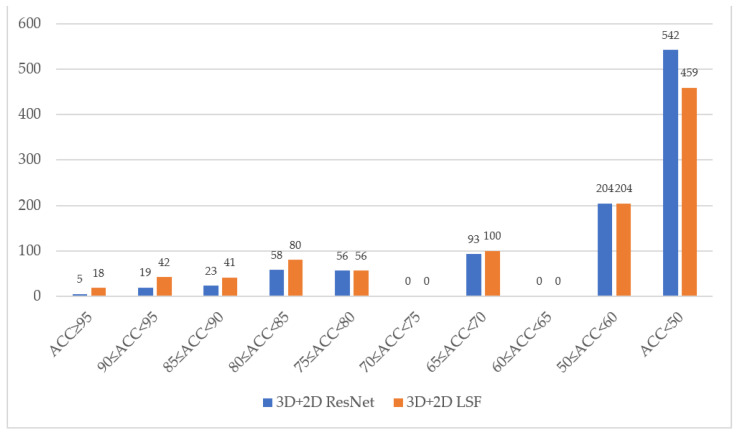
Statistics of accuracy for each class in LRW-1000. The blue histogram is the baseline model we compared, while the orange histogram is LSF.

**Table 1 sensors-22-03732-t001:** The convolution structure of different components in the front end.

Component	Convolution Structure		
	Full 3D LSF	3D + 2D LSF	Full 2D LSF
Convolution Head	3D Conv	3D Conv	2D Conv
Convolution Layer	3D Conv	2D Conv	2D Conv
Lateral Connection	3D Conv	2D Conv	2D Conv

**Table 2 sensors-22-03732-t002:** Details of LRW and LRW-1000.

	LRW [3]	LRW-1000 [15]
Source	BBC	CCTV
Language	English	Chinese
Level	Word	Word
Speakers	More than 1000	More than 2000
Classes	500	1000
Resolution	256 × 256	Multi
Head Angle	Multi	Multi
Background	Multi	Multi
Duration	1.16s	Multi
Total Samples	538,766	718,018

**Table 3 sensors-22-03732-t003:** The performance of the full 2D ResNet-18 and 3D ResNet-18 on LRW dataset. Results are reported in terms of RANK 1 identification accuracy (%).

Module	Accuracy	
	Full 2D ResNet-18	3D ResNet-18
/	80.34%	83.14%
+TSM	83.83%	83.52%
+SE-TSM	84.13%	83.60%

**Table 4 sensors-22-03732-t004:** Different data augment performance of the full 2D ResNet-18 and 3D + 2D ResNet-18 on LRW dataset. Results are reported in terms of RANK 1 identification accuracy (%).

Augment	Accuracy	
	Full 2D ResNet-18	3D + 2D ResNet-18
Cutout $\left(c=0.5\right)$	84.13%	83.60%
MixUp $\left(\alpha =0.2\right)$	80.92%	84.14%
CutMix $\left(c=0.5,\;\;\alpha =0.2\right)$	75.06%	79.11%

**Table 5 sensors-22-03732-t005:** Different structure of the front-end. Results are reported in terms of RANK 1 identification accuracy (%) on LRW.

Model Structure	Accuracy
Full 3D LSF	80.66%
Full 2D LSF	88.42%
3D + 2D LSF	88.52%

**Table 6 sensors-22-03732-t006:** Different sampling methods of LSF. Results are reported in terms of RANK 1 identification accuracy (%) on LRW.

Sampling Methods	Sampling Number (Frames)	Accuracy
Interval Sampling	1–2	88.47%
	1–3	88.52%
	1–6	87.46%
	2–6	88.14%
	1–12	87.06%
Differ Sampling	4	84.37%

**Table 7 sensors-22-03732-t007:** Different data augment performance of the full 2D ResNet-18 and 3D + 2D ResNet-18 on LRW dataset. Results are reported in terms of RANK 1 identification accuracy (%).

Fusion Methods	Accuracy	
	Full 2D LSF	3D + 2D LSF
Early Fusion	88.42%	88.52%
Late Fusion	86.64%	87.88%

**Table 8 sensors-22-03732-t008:** Compared with recently excellent works on LRW and LRW-1000 dataset. Results are reported in terms of RANK 1 identification accuracy (%).

Year	Method	Accuracy	
		LRW	LRW-1000
2019	Multi-Grained [22]	83.30%	36.90%
2019	I3D [8]	84.11%	-
2020	GLMIN [23]	84.41%	38.79%
2020	MS-TCN [5]	85.30%	41.40%
2020	DFN [24]	84.10%	41.90%
2020	CBAM [20]	85.02%	45.24%
2020	SpotFast + transformer [25]	84.40%	-
2021	TSM [6]	86.23%	44.60%
2021	BiGRU + MEM [11]	85.40% ^1^	50.82% ^1^
2021	SE-ResNet [26]	88.40%	55.70%
2022	Yang et al. [13]	88.50% ^1^	50.50% ^1^
2022	MS-TCN + MVM [12]	88.50% ^1^	53.82% ^1^
	**Ours (Full 2D LSF)**	**88.42%**	**57.70%**
	**Ours (2D + 3D LSF)**	**88.52%**	**58.17%**

^1^ Bimodal training.

**Table 9 sensors-22-03732-t009:** The top-50 accuracy of LRW. Results are reported in terms of RANK 1 identification accuracy (%).

Label	Acc	Label	Acc	Label	Acc	Label	Acc	Label	Acc
ABSOLUTELY	100%	ACCUSED	100.0%	AGREEMENT	100.0%	ALLEGATIONS	100.0%	BEFORE	100.0%
BUSINESSES	100.0%	CAMERON	100.0%	EVERYBODY	100.0%	EVIDENCE	100.0%	EXAMPLE	100.0%
FAMILY	100.0%	FOLLOWING	100.0%	INFLATION	100.0%	INFORMATION	100.0%	INQUIRY	100.0%
LEADERSHIP	100.0%	MILITARY	100.0%	OBAMA	100.0%	OFFICIALS	100.0%	OPERATION	100.0%
PARLIAMENT	100.0%	PERHAPS	100.0%	POSSIBLE	100.0%	POTENTIAL	100.0%	PRIME	100.0%
PROVIDE	100.0%	REFERENDUM	100.0%	RESPONSE	100.0%	SCOTLAND	100.0%	SERVICE	100.0%
SIGNIFICANT	100.0%	TEMPERATURES	100.0%	THEMSELVES	100.0%	WEAPONS	100.0%	WELFARE	100.0%
WESTMINSTER	100.0%	WOMEN	100.0%	MEMBERS	100.0%	PEOPLE	98.0%	POLITICIANS	98.0%
DIFFICULT	97.8%	COMMUNITY	97.8%	CUSTOMERS	97.7%	EVENING	97.7%	ECONOMY	97.7%
EDUCATION	97.7%	FINANCIAL	97.6%	CHILDREN	97.6%	CHILDREN	97.6%	REMEMBER	97.5%

## Data Availability

We trained and verified the proposed method on two large Chinese and English word-level lipreading datasets. The two datasets are the Lip Reading in the Wild (LRW) dataset and the LRW-1000 dataset corpus. The URLs of these datasets are https://www.robots.ox.ac.uk/~vgg/data/lip_reading/lrw1.html (accessed on 10 March 2017) and https://vipl.ict.ac.cn/view_database.php?id=14 (accessed on 16 October 2018), respectively.

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
