# Peer review of "Learning the Relative Dynamic Features for Word-Level Lipreading"

_sensors, 2022, doi:10.3390/s22103732_

Round 1

Reviewer 1 Report

Summary
  • Authors present an approach to visual isolated word recognition, a.k.a. word-level lip-reading.
  • The main novelty of the paper is a two stream convolutional feature extraction (front-end) network inspired by the SlowFast architecutre. Extracted features are then fed into BiGRU recurrent network in order to model speech dynamics. All videos in the dataset have the same length and the task is formulated as video classification, i.e. conceptually same as e.g. video action recognition.
  • State of the art is achieved on two popular challenging datasets: LRW and LRW-1000.
Highlights
  • The paper structure is concise and the text itself easy to read. Abstract, Introduction and Related Works clearly explain motivations, main points and differences of the paper as compared to other research.
  • The proposed model is both novel and state of the art, at least when compared on LRW and LRW-1000 datasets.
  • Authors perform ablation study so the reader can easily identify which parts of the system yield the highest improvement in accuracy.
Weaknesses
  • Some of the concepts are not explained particularly clearly. For example, Sparse Difference Sampling is difficult to understand. What is channel splicing and how does one arrive at 24 frames in the example?
  • English should be improved and there are also some typos, such as using the word "feature" instead of "figure". I recommend spelling & grammar check by a professional.
  • Authors use previously undefined abbreviations, e.g. LSF or VSR.
  • Code is not released along with the paper, which hinders reproducibility.
  • There is no attempt at applying or at least hinting how one could adopt the approach to continuous speech recognition (sentence-level lip-reading). I'm not sure how interesting single-word lip-reading is as a target application in itself.
Overall
  • Overall I did not find any major flaws in the paper and the research seems solid. I recommend accepting after some minor revisions (see Weaknesses).

Author Response

Thanks for your careful and valuable comments and we will explain your concerns points by point. Please see the attachment.

Reviewer 2 Report

The authors propose an interesting technique for lipreading. However I think the paper needs some refinements.

Please fuse sections 1 and 2, making a summary of both, and giving at the end of the new section a brief paragraph discusing the remaining of the paper (following the AIMRAD structure).

If possible, make a more summarized Abstract, giving the most important information about the contribution.

Please make a whole revision of the english, some paragraphs could be improved.

At the end of page 3, you may say Figure 1, instead of Feature 1.

At the end of page 4, you may say Figure 3, instead of Feature 3.

Relevant researchers used these structures? Please clarify why you want to say (page 5).

What does it means BiGRU? Please define al the used acronyms.

In page 6, you may say Figure 4, instead of Feature 4.

In page 7, you may say Figure 5, instead of Feature 5. Etc.

Please change sentences as: We trained and evaluated the proposed model on two VSR benchmarks. By:  The proposed model was trained and evaluated on two VSR benchmarks.

All figures and tables must be referenced into the text, please include the reference of table 1.

All equations are whole part of the text and must also respect the punctuation rules, for example the two equations in (1) stands for a comma and a dot.

Results on table 7 are very interesting, please make a more deepest discussion, since here you make a good comparative. For example, Yang et al. [26] and MS-TCN+MVM [23] give also very good results for LRW and SE-ResNet [21] for the case of LRW-1000 why to choose your proposed method?

Author Response

Thanks for your careful and valuable comments, and we will explain your concerns points by point.

Reviewer 3 Report

Dear authors,

I recommend to perform your tests against other featured methods of the state-of-art.

Author Response

Thanks for your careful and valuable comments and we will explain your concerns point.

Reviewer 4 Report

This study is described relatively specifically as a technology for word-level lipreading. However, some supplements are required for subscribers.

1) 3.2.1~3.2.3 needs to be organized into one paragraph. I don't feel the need to separate them.
2) Table 1. Is there any reason that the duration of the LRW-1000 dataset is multi?
3) In the description of Table 7, it needs to be changed to In[18]->Multi-Grained[18].
4) It is recommended to also include and discuss the results of the 50%<ACC<80% interval in Figure 7.
5) The author can review future-oriented research plans and impacts in the conclusion. This further improves the research results.

Author Response

Thanks for your careful and valuable comments and we will explain your concerns points by point.

Round 2

Reviewer 2 Report

This second version has been significantly improved.

All my concerns have been addressed.

Reviewer 4 Report

All doubts have been cleared.

In the future, the format may need to be modified for publication.